# AffordIt!: A Tool for Authoring Object Component Behavior in Virtual Reality

Sina Masnadi*          Andrés N. Vargas González†          Brian Williamson ‡          Joseph J. LaViola Jr.§

Department of Computer Science
University of Central Florida

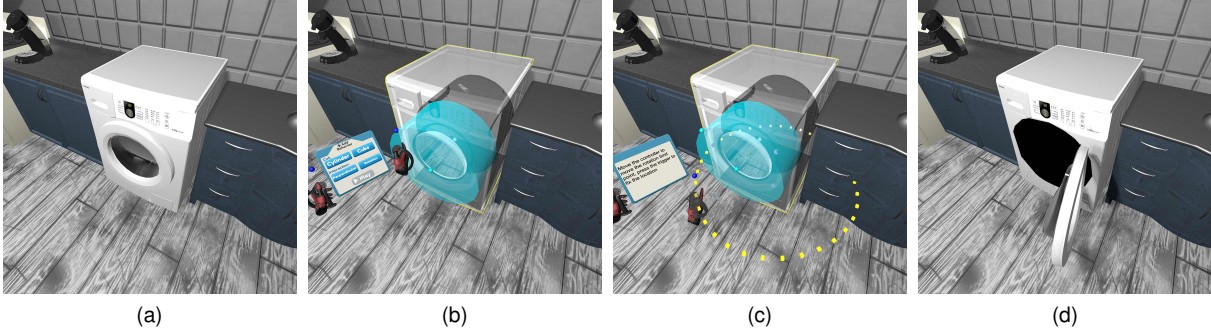

Figure 1: These figures show a sequence of steps followed to add a rotation affordance to the door of a washer machine. (a) An object in the scenario (b) Cylinder shape selection wrapping the door. (c) A user sets the amount of rotation the door will be constrained to. (d) An animation generated from the affordance can be visualized.

## ABSTRACT

In this paper we present AffordIt!, a tool for adding affordances to the component parts of a virtual object. Following 3D scene reconstruction and segmentation procedures, users find themselves with complete virtual objects, but no intrinsic behaviors have been assigned, forcing them to use unfamiliar Desktop-based 3D editing tools. AffordIt! offers an intuitive solution that allows a user to select a region of interest for the mesh cutter tool, assign an intrinsic behavior and view an animation preview of their work. To evaluate the usability and workload of AffordIt! we ran an exploratory study to gather feedback. In the study we utilize two mesh cutter shapes that select a region of interest and two movement behaviors that a user then assigns to a common household object. The results show high usability with low workload ratings, demonstrating the feasibility of AffordIt! as a valuable 3D authoring tool. Based on these initial results we also present a road-map of future work that will improve the tool in future iterations.

**Index Terms:** Human-centered computing—Human computer interaction (HCI)—Interaction paradigms—Virtual reality; Human-centered computing—Interaction design—Interaction design process and methods—Scenario-based design

## 1 INTRODUCTION

As the prevalence of virtual reality increases for simulations and video games, there is an increasing desire for the development of virtual content that is based on real scenes and environments. A problem arises when a user whose technical skills are based in

---

*e-mail: sina@knights.ucf.edu

†e-mail: andres.vargas@knights.ucf.edu

‡e-mail: brian.m.williamson@knights.ucf.edu

§e-mail: jjl@cs.ucf.edu

Graphics Interface Conference 2020
28-29 May

realistic experiences necessary to a VR scene, but not asset creation (a situation described in Hughes et al. [17]) which are needed to build a virtual scene. To alleviate this problem, recent research has been focusing on frameworks to ease users' authoring process as seen in [9, 12, 35]. 3D scene reconstruction [32, 34, 44, 49] provides a suitable solution to the problem. Initially a 3D reconstructed environment will be composed of a continuous mesh which can be segmented via autonomous tools as shown in George et al. [10] and Shamir et al.'s survey [39] or human in the loop solutions as seen in [36, 47].

However, these tools fall short at identifying and applying affordances, the intrinsic properties, of the components of the object. For example, a storage cabinet may be segmented from a larger mesh, but the movements of the cabinet door remains absent. One solution is the use of a 3D modeler, such as Autodesk Maya [33] or Blender [3], but if the user is unfamiliar with the software then a technical expert in asset creation is required. This solution carries a cost, however, as the user's own intuition and understanding of an object's affordances could be lost in translation, either in relaying requirements to a third party or to software they are not experts of. As our solution we introduce AffordIt! an online tool that allows a 3D scene author to isolate key components of virtual content and assign affordances to it using their own intuitive understanding of the object.

In this paper we define a 3D reconstructed scene as being a recreation of a real world environment that contains one or more virtual representations of an object captured within that environment. The component of an object is then defined as a segmented portion of the mesh that is not removed, but rather used to assign intrinsic behaviors. The term affordance is defined as an action that can be performed over an object (or objects) by an agent in an environment according to Gibson et al. [11]. This concept has been further expanded in the robotics field [13, 18].

AffordIt! provides an intuitive method for a scene author to select a region of interest within a continuous mesh and apply affordances to it using procedures outlined in [27, 28, 40]. Rather than relying on a sketch-based interface, we looked to the work of Hayatpur et

al. [16], in which users could invoke a plane, a ray or a point to constrain the movements of a virtual object. As such, our procedure has a user first selecting a region of interest using shape geometry followed by defining a specific movement constraint. After processing the operation on the mesh an animation demonstrates the behavior attached to it as shown in Figure 1. We evaluate this technique in an exploratory study where perceived usability and workload of the system is collected and analyzed. For the study we only use two mesh cutter geometries and two movement constraint definitions, though the concepts of AffordIt! could apply to other selection geometries or affordance definitions.

Our contributions in this paper are:

1. An interaction technique for intuitive creation of 3D shapes for mesh selection.

2. An adaptation of the Affordance template concept [11] to attach affordances to components of an object.

3. An exploratory study that analyzes how well the techniques proposed are perceived.

## 2 Related Work

There are several domains which AffordIt! touches upon. In this section we review previous research in authoring tools, geometric content creation, tools that manipulate 3D geometric constraints and smart objects in context of the internet of things.

### 2.1 VR/AR 3D Scene Authoring

The domain of 3D scene authoring is often explored in the context of scenario based authoring. In the work by Ens et al. [9] an application to author Internet of Things (IoT) devices in VR using activation links is presented, also a set of guidelines for game authoring in AR is outlined by Ng et al. in [35]. The differences between authoring in a desktop versus augmented reality is outlined in the usability study by Gonzalez et al. [12], which shows higher usability in the desktop tools than the augmented reality ones. We found Gonzalez et al.'s study to be related to ours except that they focus on individual behaviors that can be a part of a general scenario, but are not specific to the component of an object and lack the interactions AffordIt! can create. Further research into AR authoring can be seen in Lee et al. [23] that introduce the concept of an "immersive authoring" tool which uses tangible AR interaction techniques for authoring. This research allows for direct specification and testing of the content within the execution environment, similar to the results from AffordIt! that are visible at run-time. Narratives are defined in an augmented reality scene in Kapadia et al. [21]. Interestingly they create a framework for affordances that would apply well to future iterations of AffordIt!.

We also consider desktop-based authoring, which does not reflect AffordIt!, but instead show the status quo for scene development. We have found that most of this research focuses on user interactions needed when defining a tracking fiducial [24, 25, 38], such as attaching actions and behaviors to virtual content. Past research by MacIntyre et al. [25] presented many novel features for authoring an AR scene inside a MacroMedia environment, but creation does not happen at run-time. Commercial companies like ScopeAR with WorkLink [1], NGrain with Producer Pro [20], or Microsoft with Dynamics 365 Guides [31] offer solutions that ease the burden of developing complex applications, but instead allow for rapid prototyping of training experiences that can be deployed on AR devices. While we find these systems to be useful, they still rely upon asset store virtual content or 3D reconstructed content that lacks affordances.

### 2.2 Geometric Content Creation and Manipulation

Deering presented HoloSketch a novel VR sketching tool at the time to create geometric content, manipulate it and add simple animations [8]. Our work is different from HoloSketch in the interaction techniques, mesh segmentation and use context. However, different features from HoloSketch can be adapted to AffordIt!. For mesh manipulation we have found Sketch based applications to be the predominant research in this domain. SKETCH by Zeleznik et al. [50] is an early example of creating 3D objects from 2D sketches. In SKETCH constrained transformations are applied to objects, a concept that we utilize in AffordIt!. In Shao et al. [40] a sketch based application is presented that applies behaviors to concept sketches based on a region of interest selection followed an animation added to an individual part. This is similar to our approach, except that their interface is entirely 2D interactions upon a 2D object while AffordIt! explores 3D interactions and seamless visualizations with a 3D object. Commercial companies have also begun to provide a variety of tools [5, 42] that easily create 3D geometric content. AffordIt! is complimentary to these tools by providing an extension of capabilities in applying intrinsic behavior to an object.

Our interaction techniques derive from the research in object authoring by Hayatpur et al. [16] which presents three techniques for object alignment and manipulation in VR. These techniques invoke a plane, ray, or point and use hand gestures to apply movement constraints to a virtual object. Their research presents a rich set of interaction possibilities, however the idea of changing an object geometry to tie behaviors to its component parts is not studied. We address this by proposing two techniques to generate intrinsic object behaviors at run-time. First, a user is allowed to define each object component behavior from the interaction in a VR environment. Second, we apply authoring behaviors similar to [27, 28] except that we transition from a 2D sketch based interface to a 3D interaction paradigm.

### 2.3 3D Geometric Constraints

Authoring constraints has been explored in the context of objects associations based on geometries as in [43, 46]. For instance a book if placed on the top of a desk is associated to the table with one face facing the desk. In the work by Oh et al. [37], authoring objects is constrained to movements in a plane, when a collision is detected. While this is similar to our movement constraint behaviors, it is a Desktop based solution rather than authoring from within the VR environment. The theory of affordances by Gibson is divided into the concepts of attached and detached object affordances [11]. Attached objects cannot be removed from their parent object unless they become a detached one and usually have constraints in their movements. While there are successful work in robotics to apply affordance theory to provide guidelines for object manipulation [13, 19, 26], the application on 3D objects authoring is limited.

### 2.4 Smart Objects

Smart Objects are physical artifacts, enhanced with sensors and connected in a network that allows communication with humans and other artifacts as a part of the Internet of Things (IoT) paradigm by McEwen and Cassimally [30]. From an HCI perspective humans interacting with such objects face a usability challenge. Work by Matassa et al. [29] emphasize the problem of smart objects being unable to immediately communicate to people what they can afford to do. Baber et al in [2] propose a conceptual framework to exploit the affordance concept through an understanding of how humans engage with objects. The forms of engagement proposed are environmental, perceptual, morphological, motor, cultural and cognitive. As much as Internet of Things tends to lean towards a human-in-the-loop approach, the systems usability and user engagement need to be accounted for as explained by Cervantes-Solis and Baber [6]. Our approach does not fall in the IoT category but could be used as

a stepping stone to define affordances for smart objects in the IoT domain.

## 3 IMPLEMENTATION

Our technique works by first cutting a mesh using simple geometries then applying intrinsic behavior to the segmented portion. Both steps require interactions with a user to define the region of interest and the behavior. The user's interactions can be performed independent of the mesh manipulation. For the exploratory study we focused on two mesh cutter shapes and two behaviors which are defined below.

### 3.1 Mesh Cutting

For the cutting step a mesh cutter geometry is used to define the region of interest. When a cut is performed the original mesh is divided into two, one inside the mesh cutter and the other one outside. The algorithm clips the mesh using each face from the mesh cutter primitive using a brute force method as shown in Algorithm 1 in the Appendix A. The algorithm is derived from the Slicer implementation by CGAL [45] and is extended to be used on a more complex shape as the slicing tool rather than a simple plane. Triangles falling inside and outside the mesh cutter volume are segmented in two sets, while the ones who share vertices inside and outside the mesh volume are triangulated accordingly to fit in the appropriate set. The number of triangles on high polygon objects were reduced to optimize the cutting time to an order of magnitude of seconds. For more details refer to the Appendix A for a pseudocode implementation.

In the exploratory study we focused on two mesh cutters, a cuboid and a cylinder, which will be created by the user using VR controllers. Mesh cutters can be extended to any shape, a possible approach is explained in section 3.1.3. The shape geometry editing is not part of the experiment but shed some light on supporting different geometries for the mesh cutters.

#### 3.1.1 Cuboid

**Creation -** To create a cuboid in the VR environment we implemented an interaction technique that uses three points. The first two points ($P_1$) and ($P_2$) fix the corners of an initial rectangle ($R$) in 3D with a normal ($\vec{n_R}$) parallel to the floor plane (Figure 2a). Moving $P_2$ around will adjust the dimensions of the rectangle as well as its rotation over the $y$-axis. The final point ($P_3$) will define the depth of the cuboid with $R$ as its base. Let $\ell$ be the line that goes through $P_2$ and is parallel to $\vec{n_R}$, and $P_v$ as the controller's location. We will have $P_3 = proj_\ell \vec{P_v}$ which is the projection of $P_v$ on $\ell$. After creating the cuboid, it can be manipulated further to adjust its transformation matrix (Figure 2b).

**Manipulation -** In this context we define a widget as a small sphere which can be grasped by pressing a button on the controller and can be moved around in the 3D space while the button remains pressed. For rotation and translation we place a widget in the center of the cuboid that while pressed passes the rotation and translation information from the controller to the shape. Three more Widgets are placed at the defining points, $P_1$, $P_2$ and $P_3$ can be dragged to adjust the scale of the cuboid. Moving any of these widgets will fix the diagonal vertex of the cuboid in space and will scale the shape according to the position of the widget (see Figure 4a).

#### 3.1.2 Cylinder

**Creation -** A cylinder is first created by defining a line with two points ($C_1$, $C_2$) which will be the orientation and height of the shape (Figure 3a). After fixing $C_2$, the controller's location will define $P_R$ which is the closest point on the $C_2$ plane from the controller (Figure 3b). The radius of the cylinder is then calculated as $||C_2 - P_R||$. Similar to the cuboid, the transformation matrix of the cylinder can be altered after its creation.

**Manipulation -** A widget is placed in the center of the cylinder that maps the shape's rotation and translation to the rotation and

translation of the controller. A second widget is placed at $P_R$ which can be moved to alter the radius and height of the cylinder (see Figure 4b).

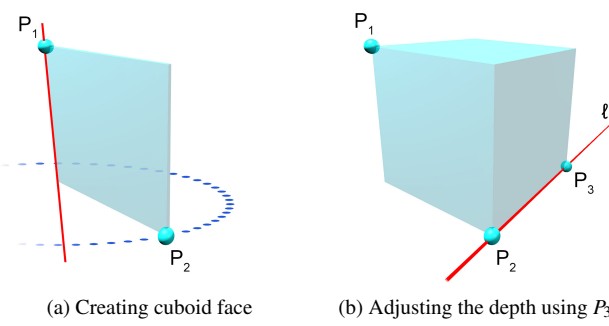

(a) Creating cuboid face     (b) Adjusting the depth using $P_3$

Figure 2: Cuboid creation

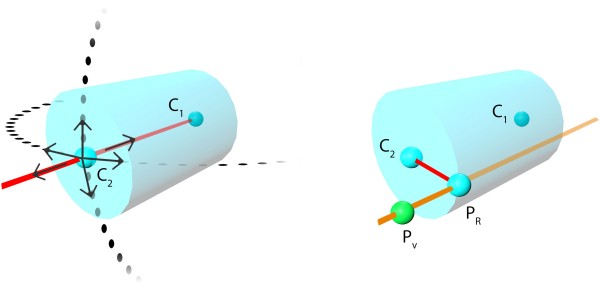

(a) Adjusting the orientation and height of the cylinder by moving $C_2$ in 3D space

(b) Adjusting the radius using projection of $P_v$ on $C_2$ plane

Figure 3: Cylinder creation

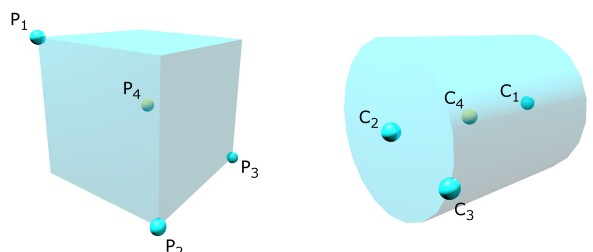

(a) $P_4$ allows cuboid translation and rotation. $P_1$, $P_2$ and $P_3$ scale the cuboid.

(b) $C_4$ allows cylinder translation and rotation. $C_1$, $C_2$ and $C_3$ allow scaling of the cylinder.

Figure 4: Manipulation widgets on primitives

#### 3.1.3 Extend to Any Geometric Shape

**Editing Geometry Mode -** The mesh cutter allows for any shape geometry to be used for segmenting an object component. Participants begin with an initial primitive which can be modified by adding

extra vertices to the mesh. As can be seen in Figure 5, on the left side of the image an extra vertex is placed in the cuboid edge by pressing and releasing the trigger button from the Vive controller on the desired position. In each vertex a widget is generated to manipulate the mesh morphology. These widgets can be dragged and the mesh changes according to the new widget position. Figure 5, shows how the newly created vertices on the right side of the image are translated upwards from the original position forming a semi-arc. While use of this feature was not a part of the study, it demonstrates how a starting primitive can be manipulated into a more complex shape.

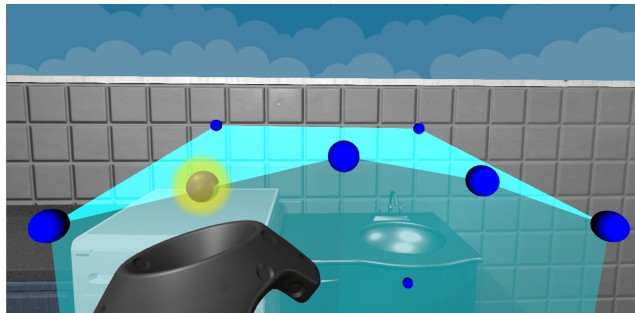

Figure 5: Highlighted in yellow a new vertex is created in editing geometry mode. Once positioned on an edge, affected faces are triangulated.

## 3.2 Interactions

An affordance we defined as part of the exploratory study was on constraining the movement of a component defined by the mesh cutter. An example of this is the key on a keyboard, which can only move in one direction (downward) and only for a fixed amount. Another example is a fridge door which can be rotated, but only around the axis of the hinge and within a specific range of angles.

We created two tools to define the movement constraints for a component based on whether it is a perpendicular or rotational interaction. Additional interactions can be implemented but a more thorough analysis on the affordance concept is required as in the work by Baber et al in [2]. Such a study falls out of the scope of this work.

### 3.2.1 Perpendicular

A perpendicular interaction is movement of an object in a straight line perpendicular to a plane (Figure 6a). This is first defined by creating three non-linear points which outline the plane perpendicular to the movement. For point placement we cast a ray from the controller to the surface of the object. Next, a grasp point ($P_g$) is placed on the object. The system automatically defines $\ell$ as the orthogonal line from $P_g$ to the plane. Finally the user defines the interaction end point ($P_e$), where the grasp location will end up after the interaction. The projection point can be moved by moving the controller, but its location is calculated by projection of the controller's location on $\ell$. Once $P_e$ is defined, the interaction is complete and the system will animate the object to demonstrate the newly defined behavior for the user.

### 3.2.2 Rotation

Rotation interactions are used for movements that are based on the rotation of an object around an axis (Figure 6b). To create this interaction the user will define the axis of rotation by placing two points ($P_1$ and $P_2$) creating a line that forms the axis of rotation ($\ell_a$). Next the grasp point ($P_g$) will be placed on the object to represent the location of effort (for example the door handle on a door). The final

point is the end trajectory point ($P_e$), which shows the location that $P_g$ will end up after rotating around $\ell_a$. This is defined by calculating the complete rotation path of a circle starting at $P_g$ and rotating around $\ell_a$. Given the controller's location $P_V$, circle center $P_c$ is calculated by $P_c = proj_{\ell_a}\overrightarrow{P_g}$. When the user presses the controller button, $P_e$ is defined using:

$$\overrightarrow{v_\ell} = proj_{\ell_a}\overrightarrow{P_v} - P_v$$
$$\overrightarrow{g_\ell} = proj_{\ell_a}\overrightarrow{P_g} - P_g \quad (1)$$
$$P_e = \overrightarrow{P_c} + ||\overrightarrow{g_\ell}||.\hat{v_\ell}$$

After fixing the location of $P_e$ we will have a complete rotation interaction and the behavior can be animated as a demonstration to the user.

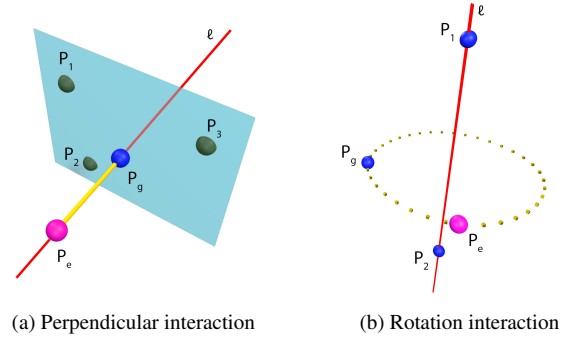

(a) Perpendicular interaction  (b) Rotation interaction

Figure 6: Interactions

## 4 USER STUDY

We performed an exploratory user study to understand the usability of AffordIt!. Post-participation surveys gathered qualitative information on usability, workload and perceived ease of use of the different aspects of the techniques. All participants used an HTC Vive Pro Eye for the study and started at the center position of a room with approximate dimensions of 4x4 meters. All virtual elements were conveniently placed so participants would not collide with real world elements during the study. We hypothesize that our tool will have high usability and low workload ratings.

### 4.1 Scenario and User Interface

The virtual scenario chosen for the experiment is a kitchen with different household appliances placed within the scene. We chose a kitchen environment so that any user can relate and have familiarity with the behavior of an appliance. Participants were allowed to interact with four objects in the scene: an oven, a washing machine, a storage cabinet and a coffee machine. Every combination of mesh cutter and affordance definition was performed on the objects. Figure 7, shows a side view of the physical area where the user study took place. The four virtual objects are super-imposed in the real room used for the study.

For the user interface we used HTC Vive Controllers as the input device. The mesh cutters and the interactions to add affordances could be invoked from a menu (see Figure 8) attached to the left hand controller with the non-dominant hand. In the same controller the track-pad button is used to show and hide the menu when pressed. For the controller on the dominant hand, a blue sphere is attached to the controller to be used as a custom pointer. The trigger button is equivalent to a "click" on a mouse and when pressed submits an action depending on the context. The gripper button when pressed

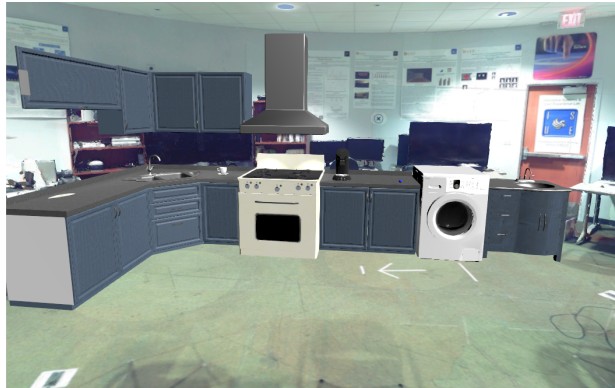

Figure 7: Side view of the 3D scanned area participants were allowed to walk. Virtual objects of interest for the study are positioned in the real world.

executes an undo. The custom pointer is used to choose an option from the Menu as shown in Figure 8 by physically hovering the button and pressing the trigger. Once an option is selected the pointer is used to place the points required to perform the operations described in the previous section.

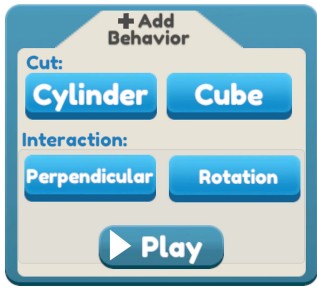

Figure 8: Menu with the different options to choose for participants.

## 4.2 Tasks

To complete the tasks participants were required to add behaviors to the objects in the scenario by invoking a mesh cutter tool (cuboid or cylinder) and define the behavior (perpendicular or rotation) of the segmented mesh.

### 4.2.1 Use a Mesh Cutter Tool to Define a Region of Interest

Participants were randomly assigned one object at a time. They decided which shape worked better to perform the object segmentation. After selecting the mesh cutter from the menu, participants approached the object and added the necessary points to create a cylinder or cube around the region of interest. If a mistake is done, the gripper button from the dominant hand controller would restart the procedure. After spawning the mesh cutter, users were allowed to transform the shape using widgets placed on the mesh geometry (see Section 3). Examples for cuboid and cylinder mesh cutters placed on objects are shown in Figure 1b and Figure 9, respectively.

### 4.2.2 Add an Interaction to the segmented part

Next, users added an interaction to the selected region by placing points following the steps defined Section 3. For each step instructions are visualized as text in the menu to help participants remember which step they are performing. For the final point, widgets are spawned to visualize the object trajectory constrained to a path (See Figure 1c). For the perpendicular interaction the path is linear and

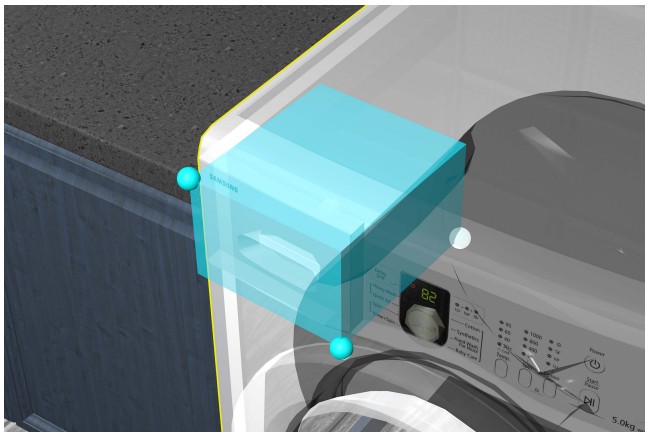

Figure 9: Cuboid mesh cutter placed on an object

Table 1: Post Questionnaire. Participants answered these questions on a 7 point Likert scale (1 = Very Little or Totally Disagree, 7 = A lot or Totally Agree).

| # | Question |
|---|---|
| Q1 | How much did the WEIGHT of the headset affected you? |
| Q2 | How ACCURATE the HTC-Vive controllers felt? |
| Q3 | How much did the PHYSICAL BUTTONS on the HTC-Vive helped with the overall experience? |
| Q4 | How much did the VIRTUAL BUTTONS on the left-hand MENU helped with the overall experience? |
| Q5 | How easy was to perform a selection of a region of interest from an object using a CUBE shape? |
| Q6 | How easy was to perform a selection of a region of interest from an object using a CYLINDER shape? |
| Q7 | How easy was to perform a ROTATION affordance around a hinge? |
| Q8 | How easy was to perform a PERPENDICULAR to a plane affordance? |
| Q9 | I enjoyed using the system overall |
| Q10 | The objects and assets in the scenario seemed realistic |

for the rotation it is circular. Users are allowed to undo one step at a time by pressing the gripper button. When the interaction is complete, the selected component will be separated from the original mesh and an animation shows the trajectory that the component is constrained to.

## 4.3 Participants and Apparatus

Sixteen people (10 male, 6 female) aged 18 to 29 ($\mu = 21.31, \sigma = 3.20$) engaged in the study. Participants were recruited from the University of Central Florida. Davis' Likert scale ratings [7] from 1 to 7, (with 1 representing not experienced or not frequently and 7 representing very experienced or very frequent) was used to measure in a pre-questionnaire the following: VR experience ($\mu = 4.00, \sigma = 1.5$), user experience with modeling toolkits & game engines ($\mu = 2.88, \sigma = 1.27$) and how frequently they played video games ($\mu = 5.75, \sigma = 1.39$). To validate the usability of the proposed techniques a VR application was developed using a HTC Vive Pro Eye headset with a resolution of 1600x1400 per eye and a field of view of 110 degrees. Two controllers were used for bi-manual interaction. Headset and controllers were tracked by HTC lighthouses. The application was implemented in Unity3D game engine using C# and SteamVR. The experiment ran on a desktop computer with an Intel Processor Core i7-8700K CPU 3.70GHz, 32 Gb RAM and a Nvidia GTX 1080Ti graphics card.

## 4.4 Study Design and Procedure

Our exploratory study was designed to be completed in approximately 45 minutes. Study participants were asked to fill out de-

mographics and pre-questionnaire forms. Next, the problem was explained for 2 minutes followed by a 5 minute video tutorial session, which allowed participants to familiarize themselves with the concepts and user interface. This was followed by a training session which was performed for an additional 5 minutes. The training session required participants to use the tools of AffordIt! following proctor instructions. An example object in the form of a modular sink with three drawers and two doors was used for training. For the experiment, participants were randomly assigned 4 different objects from the scene in Figure 10 to perform selection cuts in the objects' mesh and assign affordances to the component generated. After task completion a post-questionnaire (see Table 1) with a Likert Scale [7] from 1 (Very Little or Totally Disagree) to 7 (A lot or Totally Agree), was provided to the participant. In addition, a SUS [4] questionnaire for perceived usability of the tool and a NASA TLX questionnaire [15] for perceived workload were given to participants. Finally, participants were asked about their overall experience and any thoughts or suggestions they could have about the interface.

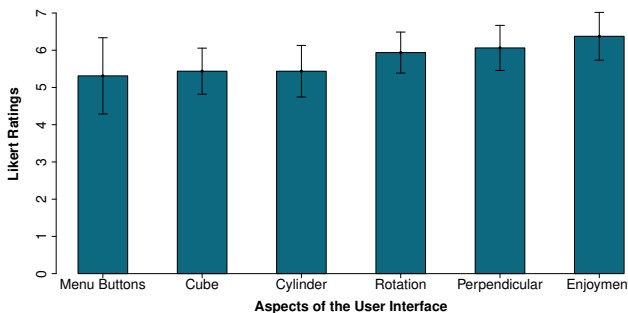

Figure 10: User study virtual environment setup

## 5 RESULTS

All participants were able to complete every task. Surveys provided to participants gathered qualitative data (Table 1) which results are shown in Figure 11. The purpose of this analysis is to identify users' scores on each individual aspect of the system, how much workload was perceived, how usable were the techniques and observations that can bring insights on future directions.

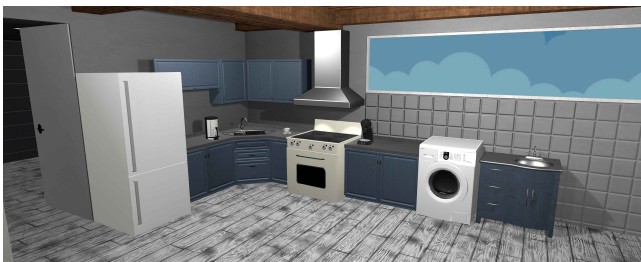

Figure 11: Plot shows the mean values and standard errors for each one of the aspects of the interface.

### 5.1 Usability and Perception

The user interface involved the use of menu buttons fixed to the left controller and placing points to define four different operations. These aspects of the interface (Q4, Q5, Q6, Q7, Q8, Q9) asked in Table 1 were rated by participants and results are shown in Figure 11. For overall usability, results from SUS scores ($\mu = 83.10, \sigma = 12.9$) show high usability for the user interface. Additionally, aspects of the hardware, such as weight of the headset, causing issues had a low

rating (Q1) ($\mu = 2.44, \sigma = 1.46$), accuracy had a high rating (Q2) ($\mu = 6.00, \sigma = 0.94$) and buttons from the controller ($\mu = 6.25, \sigma = 1.03$) were well received by participants. We conclude that these variables did not influence the correctness of the experiment. Finally we saw a high rating for the perception of realism in the environment (Q10) ($\mu = 5.88, \sigma = 0.78$).

### 5.2 Workload

Figure 12, shows scores for each subscale of an unweighted (raw) NASA TLX. A raw TLX is preferred for this study since no difference has been found in sensitivity when compared to the full version [14]. The overall subjective workload score per participant is ($\mu = 37.35, \sigma = 12.22$), which shows a low workload perception. The six factors of the NASA TLX include: Mental Demand (MD), Physical Demand (PD), Temporal Demand (TD), Own Performance (OP), Effort (EF), and Frustration Level (FL).

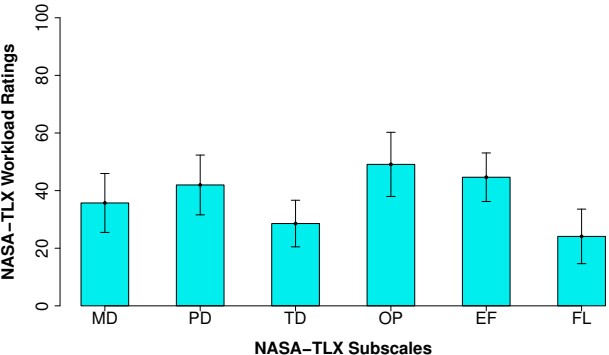

Figure 12: Plot shows the mean values and standard errors for NASA TLX workload ratings.

### 5.3 Implications

This paper evaluates the usability of AffordIt! as a tool to create behaviors in objects' components. In line with work by Hayatpur et al. [16] and Shao et al. [40], the creation and manipulation of primitives resulted in an intuitive task for participants as shown in the results. An aspect not evaluated by this work nor explored in previous work is how to extend such primitives to adapt to specific shapes that could be found in a real world scenario. This work suggests to **create or generate primitives for the purpose of selecting and segmenting mesh components**. Evaluation of how such primitives can be adjusted to specific shapes is left for future work.

The interactions presented in this work were perceived highly usable as results shown. However, based on work by Hayatpur et al. [16] and participants' comments in our study, it is suggested that **constrained movements should be authored in real-time.** This means, real-time visualization of the outcome while authoring the interaction.

Finally, the use of interactions can be extended to support more complex behaviors. In Deering [8] animation editing is conceived through components called elemental-animation objects. Following this principle, this work suggests to **implement interactions that can be easily extendable by combining them or attaching them to one or more objects.**

## 6 DISCUSSION AND OBSERVATIONS

Our exploratory study was successful in offering us several points of feedback which are discussed below.

### 6.1 Usability and Workload Analysis

In our SUS and TLX analysis we found users to rate AffordIt! as having high usability and low perceived workload. This tells us

that even this initial iteration has value in its use for affordance assignment to the components of an object. We were concerned that the virtual environment would be perceived as difficult, but the low workload rating from the TLX score assures us that users did not perceive themselves to be under a strenuous activity.

## 6.2 Post-Questionnaire Analysis

The Likert scale results from the post questionnaire provide us with additional feedback about how users felt toward the system. The low score for the headset weight (Q1) and the high score for the accuracy of the controllers (Q2) show us that the use of the HTC Vive Pro Eye did not have a negative impact on the user experience. Users found that they liked the virtual buttons (Q3) and the physical buttons (Q4). For the assigned tasks they found the creation of the cube and cylinders to be easy (Q5, Q6) and the assignment of the movement constraints to also be easy (Q7, Q8). Overall users enjoyed the system (Q9) and they found the objects and assets within the scenario to be realistic (Q10), suggesting high immersion within the scene.

## 6.3 Comment Observations

While all participants were able to create the shapes for selection and the interactions to define behaviors we found their suggestions intriguing and an avenue for opportunities for improvement.

### 6.3.1 Bring objects to the users rather than users to the objects

The study was conceived as an immersive authoring experience so the size of objects and the placement of objects within the environment replicate a real life scenario. A participant mentioned that they would prefer objects floating in the air to avoid bending to interact. We note that this is a valid point for a full VR authoring tool like in Hayatpur et al. [16].

> User: *"Sometimes I had to move my body a lot, like squating, to reach an object."*

### 6.3.2 Visual aid guidance on movement path while editing

Another intriguing set of comments was a user stating they had a good experience because of the thinking process involved while another participant did not like the outcome because of misplaced rotation points. We believe that more visual aid in the form of animations showing the movement path can help ease the thinking process of participants.

> User: *"I liked how the experiment made me think about how objects move."*

> User: *"I liked how accurate the movements were represented in VR. I disliked how sometimes the rotation points did not come out how I expected them to."*

### 6.3.3 Depth perception

Depth was perceived different among participants with the use of transparency while authoring the object behavior affected user perception of depth in some cases. A possible solution is to allow toggle transparency depending on user needs. Also outlining the edges of the shape was suggested by a participant.

> User: *"Making the meshes transparent helps with setting the location of the cylinder/box, however it makes some interactions with the object such as adding hinges difficult."*

> User: *"I liked how easy affecting objects was. I'd suggest making the textures not so transparent or emphasizing the outlines of the cube and cylinder shapes."*

### 6.3.4 Possible applications

Participants also suggested a possible use-case of AffordIt! in the following areas: game design, building interior design, education, 3D modeling programs and animations.

> User: *"useful for game design for the object interaction without coding "*

> User: *"It can be used for designing interiors or developing accurate gaming scenes with accurate animations. "*

> User: *"I think this can be useful for 3D modeling programs using VR, and for video game interactions."*

> User: *"creating a situation before actually building the real thing in irl (in real life)"*

## 7 LIMITATIONS AND FUTURE WORK

This study is exploratory in nature, and to the best of our knowledge there is no tool available for comparison at the moment. A possible baseline condition could be 3D modelers on the desktop such as Maya or Blender but the number of features and complexity would not provide a fair comparison. As so, this paper acknowledges limitations on AffordIt!, that leave room for future improvements. The study is designed as a human-in-the-loop approach, therefore inherit intuition from the users is expected to accomplish the tasks. Ideally an autonomous technique could be designed in which the geometry of the object is analyzed and a mesh cutter is designed and an affordance applied. However, we believe that the intuitive understanding of the user should be included within the process.

Segmenting the objects' parts can be done automatically through approaches such as [22, 41, 48]. AffordIt! can be used together with these tools as a human-in-the-loop tool to modify or adjust the outputs of the automatic segmentation. Intertwining the automatic approaches with AfforfIt! will provide the user an easy to use interface to correct the errors on the automatically segmented areas or use the quickly segmented areas to create affordances.

AffordIt! can be extended with more affordances and mesh cutters with a possible combination of them to produce more complex behaviors. For instance we could have an interaction that requires moving an object in a certain trajectory while rotating it at the same time, such as the behavior of a screwdriver. The mesh cutter can be extended to allow for more flexibility in shapes, for instance we could create convex polyhedrons as shown in [42]. Furthermore we intend to adopt an affordance framework as seen in Kapadia et al. [21].

Some meshes contain no internal faces, exposing a hole once the affordance is applied. We could advance our mesh cutting algorithm to also extrapolate face and normal data to the newly exposed sections of the mesh.

Also, we can develop interactions similar to [42] such that a user can draw the region of interest, snapping points to the most likely portion of the object, rather than relying on pre-defined selection shapes. This could provide increased accuracy and remove human error. As one user commented:

> User: *"Snapping surfaces of the mesh cutter to parallel surfaces of the object of interest"*

Finally, in order to provide a direct comparison to 3D modeling software, as future work we would like to conduct a larger study that seeks out modeling software experts to compare AffordIt! with traditional modeling software tools on a desktop environment. Likewise, an additional baseline condition in a desktop environment following the same principles could be implemented for direct comparison with AffordIt!.

## 8 CONCLUSION

This paper introduces AffordIt!, a set of techniques that author object components behaviors in Virtual Reality. Despite the limitations and observations found, usability results show that the interface and interaction techniques were well received by participants, as seen in the high usability scores for SUS, and had a low workload for the tasks, as shown in the low scores for TLX. Participants comments showed that they enjoyed the experience. Furthermore, the affordance techniques scored higher than the mesh cutters which can be improved, as discussed in our future work section.

There is work to be done in refining AffordIt!, but we have shown that even our initial iteration allows 3D scene authors to intuitively segment and assign affordances to an object either for scene authoring or in the development of 3D assets for a variety of use cases.

## ACKNOWLEDGMENTS

This work is supported in part by NSF Award IIS-1638060 and Army RDECOM Award W911QX13C0052. We also thank the anonymous reviewers for their insightful feedback. We are further grateful to the Interactive Systems and User Experience lab at UCF for their support.

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

# A  APPENDIX

---

**Algorithm 1** The mesh cutter algorithm

---

1: $triangles \leftarrow getObjectTriangles()$
2: $selector \leftarrow getSelectorShape()$
3: $in, out \leftarrow List()$
4: **procedure** MESHCUTTER($triangles$, $selector$, $in$, $out$)
5:     **for all** $triangle \in triangles$ **do**
6:         **if** $selector.isFullInside(triangle)$ **then**
7:             $in.Add(triangle)$
8:         **else if** $selector.isFullOutside(triangle)$ **then**
9:             $out.Add(triangle)$
10:         **else**
11:             $CutTriangle(triangle, selector, in, out)$
        **return** $in, out$
12: **procedure** CUTTRIANGLE($triangle$, $selector$, $in$, $out$)
13:     $vertsin, vertsout \leftarrow array[2]$
14:     $inCount, outCount \leftarrow 0$
15:     **for** $i = 0$ to $3$ **do**
16:         **if** $selector.isInside(triangle.vertices[i])$ **then**
17:             $vertsin[inCount] \leftarrow triangle.vertices[i]$
18:             $inCount \leftarrow inCount + 1$
19:         **else**
20:             $vertsout[outCount] \leftarrow triangle.vertices[i]$
21:             $outCount \leftarrow outCount + 1$
22:     $tmpT \leftarrow Triangle()$
23:     **if** $inCount == 1$ **then**
24:         $tmpT \leftarrow Triangle(vertsin[0], vertsout[0], vertsout[1])$
25:     **else**
26:         $tmpT \leftarrow Triangle(vertsout[0], vertsin[0], vertsin[1])$
27:     $v1, v2, v3 \leftarrow tmpT.getVertices()$
    ▷ /*getIntersectionPoint returns a point on line connecting the first two parameters where selector intersects the line*/
28:     $Pt1 \leftarrow getIntersectionPoint(v1, v2, selector)$
29:     $Pt2 \leftarrow getIntersectionPoint(v1, v3, selector)$
    ▷ /*Pt1 and Pt2 are the points where selector cut the edges of the triangles*/
30:     **if** $inCount == 1$ **then**
31:         $in.Add(Triangle(vertsin[0], Pt1, Pt2))$
32:         $out.Add(Triangle(vertsout[0], Pt1, Pt2))$
33:         $out.Add(Triangle(vertsout[0], vertsout[1], Pt2))$
34:     **else**
35:         $out.Add(Triangle(vertsout[0], Pt1, Pt2))$
36:         $in.Add(Triangle(vertsin[0], Pt1, Pt2))$
37:         $in.Add(Triangle(vertsin[0], vertsin[1], Pt2))$
    **return** $in, out$

---