# OpenReview forum: "AffordIt!: A Tool for Authoring Object Component Behavior in Virtual Reality"
_graphicsinterface.org/Graphics_Interface/2020/Conference — GI 2020_

### Official Review · AnonReviewer2 · 2020-04-20
**VR authoring tool**

**Rating:** 8
**Confidence:** 4

**Review:**

This paper is about AffordIt, a VR authoring system that helps users segment existing 3D models using geometric cutters. The system also allows users to add two basic functionalities to the objects, which are one axis rotations and translations. Finally, AffordIt shows the added functionalities as animations. The main advantage of AffordIt is that it allows users to add affordances to the objects of a scene directly in VR, which removes the need to use third-party software.

In general, I found the proposed system novel and interesting, and a good addition to the conference. The authors did a good job describing how their system works, and they discuss the limitations of their paper, which I think is important in these types of papers. My main critique of the paper is that the authors do not specify the general design decisions they took. For example, to which type of users (novices or experts), and types of products (concept vs final) they designed AffordIt. Also, the justifications for their interactions are in the related work section, which makes them difficult to follow. Clearly stating all these design decisions in their own section, will make the paper stronger.

There are also problems with the figures, which are missing some elements. This makes the description of the mesh cutting and the interactions difficult to follow. Here are specific comments:
•	Fig 3 does not have labels for Pv and nR. Also, an extra model showing how PV is projected to l might help with clarity.
•	There is also a figure missing to show the cuboid and cylinder manipulation widgets. Even if some of them are the same as P1, P2 and P3.
•	Fig 5 is not clear. It is missing an image of a 3D object with the interaction points and lines over imposed.

Finally, the description of the user study is repetitive. Especially the paragraph under section 4 title and the other sections. Also, the previous work is only referenced using the citation number. I think is better to use the system name or the author's name, so the text it's easy to follow.

Overall, this paper will be a good inclusion to the GI conference.

---

### Official Review · AnonReviewer3 · 2020-04-20
**Review of AffordIt!**

**Rating:** 6
**Confidence:** 3

**Review:**


AffordIt! is an VR authoring tool for adding translational and rotational affordances to components of an object. It consists of a mesh cutting step used to select the component and an interaction step for defining translational and rotational affordances, both of which can be done in VR without explicit coding. User study is performed to evaluate the usability and workload of the different aspects of the tool.

This paper addresses a problem that to my knowledge hasn’t been tackled. As the authors acknowledged, there are tools that could automate some of the processes like segmenting parts of an object, but there still are some benefits in enabling real-time VR authoring as algorithms may not always be available or perfect. The study results suggest that the interface and interaction techniques are relatively well-received as well.

However, the tool itself seems quite preliminary as it only allows two different geometries for mesh cutting and only translation + rotation affordances. While the study provides an evaluation of the tool and some useful insights, the study design could have been improved to obtain more useful results. For instance, some of the questions in the post questionnaire do not have any meaningful contribution and it would have been useful to report performance-related measures like completion time, breakdown of time used, or number of adjustments during segmentation.

Overall, this paper provides a preliminary solution to a problem that is of importance. Thus, I lean towards accepting it.

Minor:
-	In Figure 4, it is difficult to see which one is the new vertex. Highlighting it would be helpful.
-	Perhaps “Translation” is a better term for the “Perpendicular” interaction.
-	A picture of the study setup would be useful. Are the participants standing or sitting? How big is the VR space? (e.g., are they given the equivalent amount of space as the kitchen shown in VR?)
-	I am curious as to how participants reacted when the mesh they created didn’t perfectly overlap with the intended component of the object as it seems like there is no automatic snap function. For instance, I can imagine it being a little unattractive to perfectionists if the created mesh was noticeably larger or smaller than the washer door,
-	In Figure 1c, it is difficult to see the yellow dots demonstrating the rotation of the door. Making it bigger would be helpful. Also, I think it would be useful to visualize the door in the along with the yellow dots to preview the final rotation, which is suggested by participants.
-	Very minor, it seems a little incongruous to have a washer in the kitchen, at least to me.

Missing references:
Deering, Michael F. "HoloSketch: a virtual reality sketching/animation tool." ACM Transactions on Computer-Human Interaction (TOCHI) 2.3 (1995): 220-238.

---

### Official Review · AnonReviewer1 · 2020-04-20
**Novel tool with some weaknesses in study reporting & discussion, overall interesting work**

**Rating:** 7
**Confidence:** 3

**Review:**

AffordIt! is a tool for authoring object component behaviour within VR. With this, users can select part of a VR object, assign an animation behaviour, and preview it. The tool is a very useful and novel contribution, although I have some questions about the validity of the use case scenario. The system requires that the virtual objects are implemented in a way that they do not only present an outside facade but also contain primitives of its components not displayed on the outside (i.e., "internal faces"). This is briefly addressed in the limitations, but I would have found some discussion of this aspect very helpful, especially earlier when introducing the research motivation. How likely are designers of 3D objects to include such "internal faces"; is this common?

The paper further assessed the tool in an exploratory study looking at usability and induced workload, with promising results.
This consisted of a small user study (N=16) featuring qualitative and quantitative measures. The latter assessed usability (SUS) and workload (NASA TLX) and custom miscellaneous items.

Some issues in the study reporting:
- What was the scale range for the prior experience questions?
- The quantitative data is described as "qualitative" for some reason, even when referring to barplots in Figure 9.
- "Finally we saw a high rating for the perception of realism and feelings of immersion in the environment (Q10) (μ = 5.88, σ = 0.78)." Q10 only refers to realism - where is the immersion aspect coming from here?

For some reason, the actual qualitative aspects of the study are then reported as a subsection in the discussion (6.3 - Comment Observations). I strongly recommend that this be moved to a subsection of the previous section, i.e., the Results section.
The actual discussion of the results unfortunately is very limited (especially because large parts of it consist of qualitative reporting), and are mostly a summary, rather than a contextualization of the results within existing work, or statements on implications of the results.

The paper does discuss limitations, but I think that this section should also address the fact that the study was largely preliminary / exploratory in nature; there was no comparison condition, nor a discussion of what a baseline condition might look like for this context.

Despite these weaknesses with regards to the study reporting and discussion, the paper is interesting and showcases good and novel work and I think the GI community would benefit from its presentation (albeit with some changes as suggested above).

General minor issues:
- "users authoring process" -> "users' authoring process"

---

### Meta-Review · Area_Chair1 · 2020-04-23

**Recommendation:** Accept
**Confidence:** 4

**Metareview:**

Despite some issues (that will hopefully be largely improved for a camera-ready) and limitations, all three reviewers either tend towards acceptance or outright recommend it. I think this has the potential to spark some interesting discussions and future research at the GI conference.

---

### Decision · Program_Chairs · 2020-04-25

Accept